

# Estimating trends in atmospheric water vapor and temperature time series over Germany

Fadwa Alshawaf[1], Kyriakos Balidakis[2], Galina Dick[1], Stefan Heise[1], and Jens Wickert[1,2]

[1]German Research Centre for Geosciences GFZ, Telegrafenberg, D-14473 Potsdam, Germany.
[2]Technische Universität Berlin, Institute of Geodesy and Geoinformation Science, Straße des 17. Juni 135, 10623 Berlin, Germany.

*Correspondence to:* Fadwa Alshawaf (fadwa.alshawaf@gfz-potsdam.de)

**Abstract.** Ground-based GNSS (Global Navigation Satellite Systems) have efficiently been used since the 1990s as a meteorological observing system. Recently scientists used GNSS time series of precipitable water vapor (PWV) for climate research. In this work, we compare the temporal trends estimated from GNSS time series with those estimated from European Center for Medium-Range Weather Forecasts Reanalysis (ERA-Interim) data and meteorological measurements. We aim at evaluating climate evolution in Germany by monitoring different atmospheric variables such as temperature and PWV. PWV time series were obtained by three methods: 1) estimated from ground-based GNSS observations using the method of precise point positioning, 2) inferred from ERA-Interim reanalysis data, and 3) determined based on daily in situ measurements of temperature and relative humidity. The other relevant atmospheric parameters are available from surface measurements of meteorological stations or derived from ERA-Interim. The trends are estimated using two methods, the first applies least squares to seasonally-adjusted time series and the second using the Theil-Sen estimator. The trends estimated at 113 GNSS sites, with 10 and 19 year temporal coverage, varies between -1.5 and 2 mm/decade with standard deviations below 0.25 mm/decade. These values depend on the length and the variations of the time series. Therefore, we estimated the PWV trends using ERA-Interim and surface measurements spanning from 1991 to 2016 (26 years) at synoptic 227 stations over Germany. The former shows positive PWV trends below 0.5 mm/decade while the latter shows positive trends below 0.9 mm/decade with standard deviations below 0.03 mm/decade. The estimated PWV trends correlate with the temperature trends.

## 1 Introduction

Water vapor is considered the most active greenhouse gas that permanently affects the Earth's climate. Due to its high temporal and spatial variations, the precipitable water vapor (PWV) content in the atmosphere has to be regularly and accurately determined for meteorological and climatological purposes. PWV is the amount of water (in millimeters) that would result from condensing a column of water vapor that extends from the measurement point to altitudes of about 12 km. Water vapor mainly resides in the lowest 3 km of the atmosphere and its content generally increases with increasing air temperatures. While other observation systems such as radiosondes and microwave radiometers provide PWV measurements that are limited in the temporal and (or) spatial resolutions, ground-based GNSS provide time series of accurate PWV estimates with 15 minutes (for this work) sampling at dense GNSS networks, without significant additional costs. Since *Bevis et al.* (1992) presented the





Global Positioning System (GPS) as an efficient meteorological tool, GNSS data have been increasingly used for estimating atmospheric parameters, particularly precipitable water vapor (*Gendt et al.*, 2004; *Luo et al.*, 2008; *Jade and Vijayan*, 2008; *Bender et al.*, 2008; *Alshawaf et al.*, 2015). GNSS-based estimates of Zenith Total Delay (ZTD) or PWV have been used to improve numerical weather prediction (NWP) models (Bock et al., 2005; *Bennitt and Jupp*, 2012). They have also been

used to improve the performance of high-resolution atmospheric models (*Pichelli et al.*, 2010). Besides meteorology, GNSS estimates of PWV have been employed over Scandinavia for Climatological Research (*Elgered and Jarlemark*, 1998; *Gradinarsky et al.*, 2002; *Nilsson and Elgered*, 2008) . The authors found that PWV shows an increase of 1.2–2.4 mm per decade. *Haas et al.* (2003) used ground-based GPS, very long baseline interferometry, radiosonde, and microwave radiometer data to assess long-term trends in PWV time series over Sweden. An increase of about 0.17 mm/year within the period 1980–2002

was observed. *Hausmann et al.* (2017) analyzed a decadal time series of PWV (2005–2015) from mid-infrared FTIR ( Fourier transform infrared) measurements above mountain Zugspitze. For that time period, they did not observe statistically significant trend in PWV time series. The PWV time series from ground-based GNSS and the European Center for Medium-Range Weather Forecasts (ECMWF) Reanalysis (ERA-Interim) data might show temporal inconsistencies due to, for example, hardware replacement or inconsistent processing methods (*Ning et al.*, 2016). Therefore, homogenization of the atmospheric data is

indispensable for climatological research to properly estimate climatic long-term trends. *Vey et al.* (2009) and *Ning et al.* (2016) analyzed PWV time series estimated at global GNSS sites to detect and correct for inhomogeneities in the data. Atmospheric reanalysis models such as ERA-Interim have also been employed for climate research. The analysis fields are produced based on 4D-Var assimilation of regular and irregular meteorological data, including surface and upper-air atmospheric fields (*Dee et al.*, 2011).

*Bengtsson et al.* (2004) observed an increasing long-term trend with a slope of 0.16 mm per decade in the water vapor data set of ERA 40 within the period of 1958–2001. They suggested to apply corrections for the changes in the observing system when using the data for PWV analysis to achieve trend values comparable to GNSS. ERA-Interim and MERRA (Modern Era Retrospective-Analysis for Research and Applications) where also used for trend analysis (*Simmons et al.*, 2007; *Suarez et al.*, 2008).

Typically, climate scientists consider a period of 30 years as an appropriate time over which to average variations in weather and evaluate climatic effects for a particular site, as described by the World Meteorological Organization (*Arguez and Vose*, 2011). Data collected and averaged or summed in some way over 30 years are referred to as climate normals. A 30 year period is recommended, as it is sufficiently long to filter out the interannual variations or anomalies, but at the same time short enough to show climatic trends. It is then obvious that the GNSS temporal span is still too short for estimating a reasonably proper climatic

trends in this sense. The previous studies using GNSS-based PWV time series for assessing the trends show highly variable estimates for different time windows as well as different research regions. In this paper, we present the PWV trends estimated using GNSS sites over Germany and compare them with the trends estimated from other data sets. The current climate normal period should cover the period from 1 January 1991 to 31 December 2020 (*Arguez and Vose*, 2011). So, we analyzed time series available from 1 January 1991 to June 2016 (26 years) to provide more robust information about the climatic trends. These data

sets are the ERA-Interim reanalysis and surface meteorological data from the German Meteorological Service (DWD). The





former data set provides global PWV grids while the latter does not. However, different studies have used the dew point that is computed using surface measurements of temperature and relative humidity to approximate the total column PWV (*Reitan*, 1963; *Bolsenga*, 1965; *Smith*, 1966; *Tuller*, 1977). The formula presented to obtain the PWV from surface measurements are described in section 3. This empirical relation requires only information that can accurately be determined on the ground. The

accuracy of dew point-based PWV approximations depends of course on the atmospheric conditions and the variability of the moisture profiles. It is however obvious that PWV estimations based just on atmospheric conditions at the Earth's surface would not always be in complete agreement with, for example, PWV values from balloon soundings integrated through the atmosphere. Since the possibility for obtaining a data set with long time series and high spatial resolution for estimating PWV trends is very limited, we evaluated the potential of this method for climate analysis. We first obtained the PWV based on dew

point temperature measurements and evaluated the quality of the time series. Then we used them to estimate the PWV as well as temperature trends. In this work, we apply a preprocessing step to evaluate the quality and homogeneity of the time series ahead of the trend estimation. For checking the homogeneity of the time series, we use the ERA-Interim as a reference data set. We apply the technique of the singular spectrum analysis to detect possible change points followed by t-test to identify the significance of them. The description of the approach for homogeneity check is beyond the scope of this paper and details are

found in (*Wang*, 2008; *Ning et al.*, 2016). In the following, we will present the results of comparing PWV from different data sets and the trend analysis.

This paper is organized as follows. In section 2, we describe the method for PWV determination using GNSS data and the comparison with ERA-Interim. The method to obtain PWV based on surface measurements of temperature and relative humidity is described in section 3. In section 4, we present the methods for estimating the atmospheric trends followed by the

results of estimating the decadal rate of change in section 5. The conclusions of this research are ultimately presented.

## 2   Determination of atmospheric PWV from GNSS data

We used GPS data collected in central Europe, mainly in Germany as shown in Figure 1. The research region is well covered by 351 permanent GNSS sites. Homogeneous time series with length from 10 to 19 years are available from 119 sites. Besides GNSS, there are 326 meteorological stations operated by the German Meteorological Service (DWD) with data profiles span-

ning more than 60 years at a temporal rate of 1 hour. The climate data center created by the DWD provides long homogeneous time series for climate studies (http://www.dwd.de/EN/climate_environment/cdc/cdc_node.html). They provide surface measurements of temperature, pressure, water vapor pressure, precipitation, snow cover and other meteorological parameters for climate research. We also used the ERA-Interim reanalysis data with a spatial resolution of 79 km in longitude and latitude directions and 6 hours temporal resolution. In this section, we briefly describe the methods for PWV determination using GNSS

phase observations and a comparison between the different data sets.

Based on the method of precise point positioning (*Zumberge et al.*, 1997), GNSS observations are processed to produce site-specific atmospheric ZTD. The ZTD is an estimate of the total propagation delay caused by the dry gases and water vapor of the atmosphere. Employing meteorological data measured directly at the GNSS site or interpolated from the adjacent





meteorological station, the zenith hydrostatic delay (ZHD) is calculated. For each GNSS site, the nearest meteorological station triangle is used to interpolate the measurements at that site (*Gendt et al.*, 2004). The ZHD, in meters, at the GNSS site is then calculated according to the model of *Saastamoinen* (1973) reported in (*Davis*, 1986, pp. 51):

$$ZHD = \frac{0.002277P}{1 - 0.0026\cos 2\phi - 0.00028H} \tag{1}$$

where $H$ is the orthometric height in km and $\phi$ is the latitude of station. $P$ is the corresponding air pressure at the station in hPa. The air pressure $P$ at the GNSS site in Eq. 1 is obtained by vertically interpolating the surface pressure $P_s$ using the barometric formula:

$$P = P_s \left( \frac{T_s - L(z - z_s)}{T_s} \right)^{\frac{gM}{RL}} \tag{2}$$

where $T_s$ is the surface air temperature at the meteorological station in [K], $z$ and $z_s$ are respectively the altitude in [km] of the GNSS and meteorological station above mean sea level (AMSL), $L$ is the temperature lapse rate in [K/km], $R$ is the universal gas constant (8.31447 J/mol K), $M$ is the molar mass of Earth's air (0.0289644 kg/mol), and $g$ is the Earth's gravitational acceleration (9.80665 m/s$^2$). The temperature is related to the elevation change using the following linear regression:

$$T = T_s - L(z - z_s) \tag{3}$$

By analyzing ERA-Interim temperature profiles over Germany, we found that the Lapse rate changes between summer and winter and in space. The value of $L$ varies between 3 and 7 K/km for this research region. These values result in 2 mm change in the ZHD at altitude difference of 1 km. Similarly, the change in PWV is below 0.2 mm, which can be neglected. Once the ZHD is calculated, the zenith wet delay (ZWD) is obtained by:

$$ZWD = ZTD - ZHD \tag{4}$$

and it is converted into PWV using the empirical factor $\Pi$ (*Bevis et al.*, 1994),

$$PWV = \Pi \cdot ZWD \tag{5}$$

For more details on the GNSS data processing, the reader is referred to (*Gendt et al.*, 2004) and (*Bender et al.*, 2011).

We compared the PWV obtained from GNSS with ERA-Interim data. Figure 2 shows the results for three sites at different altitudes as well as the mean and standard deviations of the time series difference. ERA-Interim grid provides values of PWV at grid points separated by about 79 km in longitude and latitude. The ECMWF provides a software to horizontally interpolate the current ERA-Interim grid at different locations of the GNSS stations as described in (*Heise et al.*, 2009). We did not account for altitude difference, which have significant impact in mountainous areas. For the sites located in flat terrain, the two data sets show strong correlation with a mean difference below 1 mm and uncertainty values of less than 2 mm (Figure 2). The mean difference increases for sites in mountainous regions. The time series of the site 0285 (Garmisch, Germany, 1779 m AMSL),



30  for example, show a larger bias between GNSS and ERA-Interim data, which is explained as follows: we average PWV of four distant grid points around the GNSS site. With the rough spatial resolution, the variability of surface topography is not well captured in the reanalysis data, which significantly increases the height difference between GNSS and the model, and hence the PWV difference. Besides, the daily mean in ERA-Interim is obtained by averaging four PWV values/day, while using GNSS there are 96 PWV estimates/day. We should bear in mind that GNSS estimates of PWV in mountainous regions might be less accurate because of shadowing effects. Due to the presence of mountains, the visibility of satellites might be limited. Also,

there might be multipath effects in the observed signal. This will have an impact on the estimated tropospheric parameters.

For accurate determination of the PWV from GNSS measurements, it is required to have measurements of mainly air pressure and temperature at the GNSS sites or within a short spatial range. In the absence of meteorological measurements, would the interpolation of pressure and temperature from reanalysis data be a good replacement? To answer this question, we compared the PWV time series extracted from the ZTD by using both measurements at the meteorological stations and ERA-

Interim data. To calculate the ZHD, the in situ measured pressure and temperature are horizontally interpolated to the GNSS site and then vertically interpolated to the GNSS antenna phase center. For GNSS sites below the lowest ERA-Interim level, the pressure and temperature are extrapolated at the site altitude as described in (*Heise et al.*, 2009). The ZWD is then extracted and converted into PWV. Figure 3 shows the scatterplots of PWV obtained using surface measurements and ERA-Interim data. We found that in regions of smooth topography, the ERA-Interim data and the measurements provide almost the same values

of PWV and pressure. In regions of steep topographic gradients, however, the ERA-Interim data show slightly different results, which is mainly related to the pressure data as observed from Figure 3. The deviations between the measured pressure and the ERA-Interim pressure increase in mountainous regions, which affects the calculation of the ZHD and hence the obtained PWV.

Besides station pressure, an important factor for an accurate determination of PWV is the conversion factor $\Pi$, which should be calculated using measurements of surface temperature. *Askne and Nordius* (1987) determined the conversion factor $\Pi$ as

follows:

$$\Pi = \frac{10^6}{\rho_w R_w \left( \dfrac{k_3}{T_m} + k_2' \right)} \tag{6}$$

where $\rho_w$ is the density of water and $R_w$ is the specific gas constant of water vapor [461.5 J/kg·K]. In our research, we used the values of the physical constants $k_3$ and $k_2'$ given by *Bevis et al.* (1994), $T_m$ was given by *Davis et al.* (1985) as

$$T_m = \frac{\int_z \frac{P_{wv}}{T} dz}{\int_z \frac{P_{wv}}{T^2} dz} \tag{7}$$

where $T$ is the air temperature and $P_{wv}$ water vapor pressure at vertical levels. *Davis et al.* (1985) suggested the use of water vapor pressure and temperature profiles from radiosondes; however, it is easier to get these profiles from numerical atmospheric models. In this work, we obtained $T_m$ as described in (*Heise et al.*, 2009) using the ERA-Interim model that covers 60 vertical levels extending from the Earth's surface up to 0.1 hPa. $T_m$ can be well approximated based on air surface temperature by the





following formula (*Bevis et al.*, 1992):

$$T_m \approx 70.2 + 0.72 T_s \qquad (8)$$

$T_s$ is the surface temperature in [K]. For our research region, we compared $T_m$ obtained from both methods (7) and (8) as shown by the scatterplot of Figure 4. The surface temperature and vertical profiles of water vapor pressure and temperature in Eq. 7 from ERA-Interim were employed. The difference between the $T_m$ calculated from both methods at the GNSS site 0522 (Pirmasens, Germany, 399 m AMSL) has a mean value of 0.97 K and a standard deviation (STD) of 2 K. Repeating the calculations for the site 0285 (Garmisch, Germany, 1779 m AMSL), the mean difference increases to 3.02 K and the STD is 1.83 K. Not only surface pressure grids are inaccurate in mountainous regions (Figure 3 d), but also pressure profiles, which might be related to the coarse grid of ERA-Interim. Also, the temperature profiles have inaccuracies, however, less than those for the pressure. By using the integration in Eq. 7, the accumulated error in the calculated $T_m$ will be higher; and the bias between this $T_m$ and that calculated using only the surface temperature will increase, as observed from the right plot in Figure 4. However, by computing the PWV using the two different values of $T_m$, the results show a mean difference of 0.048 mm for site 0522 and -0.083 mm for site 0285. Hence, Eq. 8 will be used to calculate $T_m$ since it only requires the measured surface temperature.

## 3 Determination of PWV based on surface meteorological measurements

It is not possible to accurately determine the total column water vapor using surface meteorological observations alone. However, it was shown in the 1960s that it is possible to approximate the atmospheric PWV based on dew point temperature measurements, which is considered an indicator of the amount of moisture in the air (*Reitan*, 1963). The dew point temperature in turn is determined based on the air temperature and relative humidity. *Reitan* (1963) presented a basic relationship between the mean monthly PWV and mean monthly surface dew point temperature by the following regression form:

$$PWV = \exp(b T_d + a) \qquad (9)$$

where $PWV$ is in cm and $T_d$ is the dew point temperature in °F. $a$ and $b$ are estimated to have the values of -0.981 and 0.0341 (*Reitan*, 1963). The standard error in the PWV estimate was 0.18 cm. Following the same procedure, *Bolsenga* (1965) obtained slightly different estimates for $a$ and $b$ using hourly and mean daily observations. *Smith* (1966) obtained a similar regression equation with the coefficient $a$ not being constant. It rather depends on the vertical distribution of the atmospheric moisture, i.e.,

$$PWV = \exp\left(\underbrace{0.0393}_{b} T_d + \underbrace{[0.1133 - \ln(\lambda + 1)]}_{a}\right) \qquad (10)$$

with the value of $\lambda$ dependent on the site latitude and the season of year (*Smith*, 1966). The surface measurements of relative humidity are necessary to determine the dew point temperature $T_d$, which can be related as presented by *Lawrence* (2005)





using the following formula:

$$T_d = \frac{B_1 \left[ \ln \frac{rh}{100} + \frac{A_1 T}{B1 + T} \right]}{A_1 - \ln \frac{rh}{100} - \frac{A_1 T}{B1 + T}} \tag{11}$$

where $rh$ is the relative humidity in percentage and $T$ is the surface air temperature. Both $T$ and $T_d$ are given in degrees Celsius. The coefficients $A_1$ and $B_1$ have the values 17.625 and 243.04 °C, respectively.

In this work, we estimated the coefficients $a$ and $b$ at each meteorological station by fitting the curves in Eq. 9 to the ERA-Interim PWV data. Figure 5 (a) shows an example of the fitting at station Lindenberg and the estimated $a$ and $b$ at 227 stations in (b). The estimated $a$ values tend to show higher variability at higher altitudes (above 700 m), while the coefficient $b$ shows lower change with the altitude. The median values for $a$ and $b$ using daily PWV are -1.346 and 0.039, which are close to the values -1.249 and 0.0427 presented by *Bolsenga* (1965). For monthly PWV, the median values are -1.224 and 0.037 for $a$ and

$b$, respectively.

To evaluate this method, we used the temperature and relative humidity measurements at the meteorological station Lindenberg (14°6'E, 52°12'N) to determine the PWV using Eq. 9. We compared the obtained PWV values with the radiosonde measurements of PWV at 12:00 UTC, as shown in Figure 6. The time series have a 91% correlation, a mean difference of 0.04 mm and a difference STD of 3.2 mm. To evaluate the daily PWV time series for the whole network, we used the ERA-

Interim data. The PWV value at the meteorological station is computed by applying bilinear interpolation to ERA-Interim PWV at four grid points around that station. The altitude difference was not accounted for. Figure 7a shows the bias and standard deviation values of daily PWV for 227 stations as well as the mean difference against the altitude difference of the two data sets (ERA-Interim height−station height). The mean difference is centered around 0.15 mm and the standard deviation around 2.5 mm. From Figure 7b we observe that the higher the altitude difference, the larger is the mean PWV difference.

## 4    Decadal variability in time series of atmospheric variables

### 4.1    Estimating the trend using least squares regression

Econometricians developed reasonably simple models that are capable of interpreting, testing hypotheses, and forecasting economic data. The method was to decompose the time series into a trend, a seasonal, a cyclic, and an irregular component (*Enders*, 1995). The trend component represents the long-term behavior of the time series, while the seasonal and the cyclic

components represent the regular and periodic movements. The time series also contain a stochastic irregular component. Time series of PWV and temperature, for example, have different temporal variations that can be reasonably modeled using this approach. Here holds an additive model, such that the time series $y_t$ can be extended as:

$$y_t = T_t + S_t + I_t \tag{12}$$

where $T_t$ is a deterministic trend component with slow temporal variations, $S_t$ represents the seasonal component with known

periodicity (e.g., 12 months for PWV and temperature), and $I_t$ represents the irregular (stationary) stochastic component with




short temporal variations. We did not observe a regular signal that lasts longer than one year, so we excluded the cyclic component for the model. The presence of seasonality might mask the small changes in the linear trend. Therefore, for proper trend analysis, the seasonal component has to be estimated and removed from the time series, which is known by seasonal adjustment (*Enders*, 1995). The deseasonalized data are useful for extracting the long-term trend and exploring the irregular component of a time series.

The seasonal adjustment is applied as an iterative procedure as follows. To best estimate the seasonal component, the linear
trend has first to be estimated and removed from the time series. There are different methods to estimate the trend such as using moving average or parametric trend estimation. Here, we used the method of moving average with a window length of one year that is able to smooth out seasonal and irregular signals. We employ time series of PWV and temperature with daily values (the GNSS-based estimates of PWV have a temporal resolution of 15 minutes, but we average them to get mean daily values for climatological studies). The trend is estimated as follows:

$$\hat{T}_t = \frac{y_{t-q} + y_{t-q+1} + \cdots + y_{t+q-1} + y_{t+q}}{d} \tag{13}$$

Since the time series are daily and the seasonal signal is annual, the value of $d$ is 365 and $q = (d-1)/2$. For $d = 366$, $q = d/2$ and the trend is estimated from:

$$\hat{T}_t = \frac{0.5y_{t-q} + y_{t-q+1} + \cdots + y_{t+q-1} + 0.5y_{t+q}}{d} \tag{14}$$

The estimated trend component is subtracted from the original time series and the detrended signal is averaged to estimate
the seasonal component $\hat{S}_t$ as follows. We first obtain:

$$w_t = \frac{1}{\text{number of summands}} \sum_{\frac{q-t}{d}}^{\frac{n-q-t}{d}} \left( y_{t+jd} - \hat{T}_{t+jd} \right) \tag{15}$$

with $n$ the number of data samples. Then $w_t$ is centered, i.e., we derive a seasonal signal with a zero mean.

$$\hat{S}_t = w_t - \frac{1}{d} \sum_{k=1}^{d} w_k, \qquad t = 1, 2, \cdots, d \tag{16}$$

For an additive model, $\hat{S}_t$ should fluctuate around zero to avoid any influence from the trend. The estimated seasonal component
is subtracted from the original time series to obtain a seasonally-adjusted time series $dy_t$, i.e.,

$$dy_t = y_t - \hat{S}_t \tag{17}$$

Figure 8 shows an example of the trend, seasonal, and irregular components of PWV time series at site 0896 in Berlin. To estimate the slope of the trend, we fit a straight line $\hat{T} = \hat{b} + \hat{m}\,t$ to the trend component produced by the moving average step. The standard deviation of the estimated slope (called standard error) is calculated as (*Wigley et al.*, 2006):

$$s_{\hat{m}}^2 = \frac{\frac{1}{n-2} \sum_1^n (y_i - \hat{y})^2}{\sum_1^n (t_i - \bar{t})^2} \tag{18}$$



where $n-2$ is the degree of freedom for $n$ data points. The approximate 95% confidence interval is expressed as $\hat{m} \pm 2\,s_{\hat{m}}$. *Weatherhead et al.* (1998) presented another way to calculate the standard deviation of the estimated slope.

$$s_{\hat{m}}^* = \frac{\sigma_I}{n_y^{3/2}} \sqrt{\frac{1+\phi_I}{1-\phi_I}} \tag{19}$$

where $\sigma_I$ denotes the standard deviation of the irregular component and $n_y$ denotes the number of years of the data. $\phi_I$ represents the 1-Lag autocorrelation of the irregular component.

## 4.2 Estimating the trend using Theil-Sen estimator

The Theil-Sen estimator presented by *Theil* (1950) and *Sen* (1968) aims to robustly find the linear fit of a data set despite containing outliers. If $(t_1, y_1), \ldots, (t_n, y_n)$ represent the data points, then the Theil-Sen estimator determines the slope of the line that connects each data pair. The median among the slopes of all pairs in the slope of the fit, i.e.,

$$\hat{m} = \text{median}\left\{ \frac{y_j - y_i}{t_j - t_i} \right\} \qquad \text{for } i < j \leq n \tag{20}$$

The standard error of the estimated slope is calculated as in (18).

We compared the two methods of trend estimation using PWV time series at the site Lindenberg (14°6'E, 52°12'N), where GNSS, ERA-Interim, synoptic and radiosonde data are available Figure 9. The mean difference between PWV from synoptic data and GNSS is 0.04 mm, while their to the ERA-Interim is -0.21 mm. The mean difference of both GNSS and synoptic PWV to the radiosonde PWV is 0.95 mm, which is because the former are daily values while the radiosonde provides an instant measurement (at 12:00 UTC). This, however, does marginally affect the estimation of the trend. For the least squares method, we estimate and remove the seasonal component and filter out the irregular component to provide the trend shown in Figure 9 (b). Table 1 shows the slope of the linear trend estimated at the site Lindenberg using the PWV time series in Figure 9 computed using the least squares and Theil-Sen methods. Applying both methods to three different data sets shows a positive trend of about 0.5 mm/decade with standard deviation of 0.04 mm/decade.

## 5 Results

In this section, we show the estimated trends using three data sets, GNSS, ERA-Interim, and synoptic data of PWV and temperature. First, we estimated the trends of PWV at 351 GNSS sites with time series of length 4 to 19 years and the corresponding standard deviations of the estimated slope as shown in Figure 10 (a, b). The size of the marker is proportional to the length of the time series (small squares indicate short time series and larger ones indicate longer time series). As observed from the figure, there are high trend values, particularly at sites with short time series. Therefore, in Figure 10 (c, d), we eliminated all sites with time series shorter than 10 years. At the remaining 119 sites the PWV trend varies between -1.5 to 2 mm/decade (except for six sites) with precision of the estimated trends below 0.2 mm/decade.





In order to get more insight and more reasonable conclusions about the long-term temporal variations of PWV, it is necessary to analyze longer time series. Since the last climate normal extends from 1991 to 2020, we analyzed time series of 26 years (January, 1991–June, 2016) from ERA-Interim and synoptic data. We investigated time series at 227 meteorological stations where the ERA-Interim is horizontally interpolated at the synoptic station using bilinear interpolation. Figure 11 shows the estimated trends using ERA-Interim PWV time series by applying, first the least squares to the seasonally-adjusted data and second using the Theil-Sen method. Both methods show similar values of the trend, positve and below 0.5 mm/decade. The standard deviations (errors) of the estimated trends using the Theil-Sen method are higher. As observed from the Figure 11, the trend tends to increase in the direction to northeastern Germany.

Using the method of *Reitan* (1963), the PWV can be obtained based on surface measurements of dew point temperature. The potential of this data set is the length of the time series that might go back to the beginning of the twentieth century. The DWD checks the quality and homogeneity and provides time series of atmospheric parameters that are proper for climate studies. We used the surface measurements of temperature and relative humidity to obtain the PWV and compared the results with the ERA-Interim before the trend estimation. The difference between the data sets is displayed in Figure 7. Next, we estimated the trends using the time series of PWV, which are presented in Figure 12. The trends estimated at the sites in the northeastern part of Germany show higher values; however, we do not observe the gradient, with the same consistency, shown by the ERA-Interim data, for example, in Figure 11 (a). Considering these differences, we have to keep in mind that the synoptic data are point measurements that are affected by the local environment (surroundings) of the meteorological station and weather conditions. Also, the parameters $a$ and $b$ in Eq. 9 might result in slight spatial variations. Such small scale effects can not be represented by the ERA-Interim data due to the coarse spatial resolution. Therefore, we attempted to reduce these small scale effects from the data by curve fitting, where a cubic polynomial is applied along the longitude and latitude. Figure 13 shows an example of the PWV data at 227 stations and the fitted PWV. The fitting is applied in the longitude direction and the fitted PWV is subtracted from the original data and the fitting is applied to the residuals along the latitude. The final fitted PWV is the sum of both fittings along the longitude and latitude. Applying the 1D polynomial regression sequentially over the longitude and latitude leads to better fitting than applying 2D polynomial to the data in longitude and latitude. Figure 14 shows the trends estimated using the filtered data set. In this figure, we observe the increase in the estimated trend when moving towards northeastern Germany. The color gradient in this figures is similar to that shown by ERA-Interim in Figure 11. However, the values of the slopes estimated from ERA-Interim and synoptic data are different, which is not surprising. First because of the coarse resolution of the ERA-Interim data and second due to altitude difference, which might result in different trends. In order to justify these results, a data set with a higher spatial resolution than that of ERA-Interim is required.

The same procedure is applied to the temperature time series. The estimated temperature trends from surface measurements at 227 stations are shown in Figure 15. We observed that the estimated trend in PWV is correlated with that from the temperature, which is exhibited in Figure 16. Using ERA-Interim temperature and dew point temperature leads to the same observation; however the trend values are slightly different. We also observed that the trends of dew point temperature are almost in the same range as those for PWV, which makes time series of the dew point temperature proper to provide reasonably adequate information about PWV trends.





# 6 Conclusions

In this paper, we aimed at analyzing the climate evolution in Germany using time series of precipitable water vapor and surface air temperature. We first compared PWV time series obtained from GNSS, ERA-Interim, and synoptic observations to check the quality thereof. The data sets show strong correlation with uncertainty values below 1 mm.

By comparing the GNSS-based PWV with those from ERA-Interim, the results show strong agreement in flat terrain while a bias of about 0.6 mm is observed in mountainous regions. This might be due to the coarse resolution of the ERA-Interim data and hence the lack to properly represent the topography. In the absence of pressure and temperature measurements, ERA-Interim data is an appropriate replacement for extracting PWV from the GNSS path delay, particularly in flat terrain.

Considering long term evolution, PWV trends have a physical meaning only when the time series have to be adequately long. Therefore, we used time series from ERA-Interim and synoptic stations. Using dew point temperature, we could produce PWV time series at 227 stations with an average bias below 1.2 mm to the ERA-Interim data. To evaluate the temporal evolution of PWV and temperature, we modeled the time series with an additive model that contains trend, seasonal, and stochastic irregular components. The time series are seasonally adjusted to remove the periodic signal and the trend component is then analyzed after filtering out the irregular component caused mainly by weather variations. The Comparison of this method with the Theil-Sen estimator shows insignificant differences. The GNSS-based estimated PWV trends change between -1.5 and 2 mm/decade for time series that are 10 to 19 years long. By analyzing time series of 26 years from ERA-Interim and filtered synoptic data the trends are observed to be positive and below 0.9 mm/decade. The ERA-Interim PWV shows lower trend values of the trend. We found that the trend tends to show a positive gradient when moving from southwestern to northeastern Germany. The PWV trends correlate with the trends of the temperature while the magnitude of the PWV trend slightly differs from that of the dew point temperature. Hence, we can consider the trends estimated from the dew point temperature as a measure for the PWV trends in case of lack of observations.

It would be illuminating to validate the results of this research using a data set that has a higher spatial resolution than the ERA-Interim.

*Acknowledgements.* The authors would like to thank the ECMWF for making publicly available the ERA-Interim data. Thanks also go to the German Meteorological Service (DWD) for providing us with hourly meteorological measurements.



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

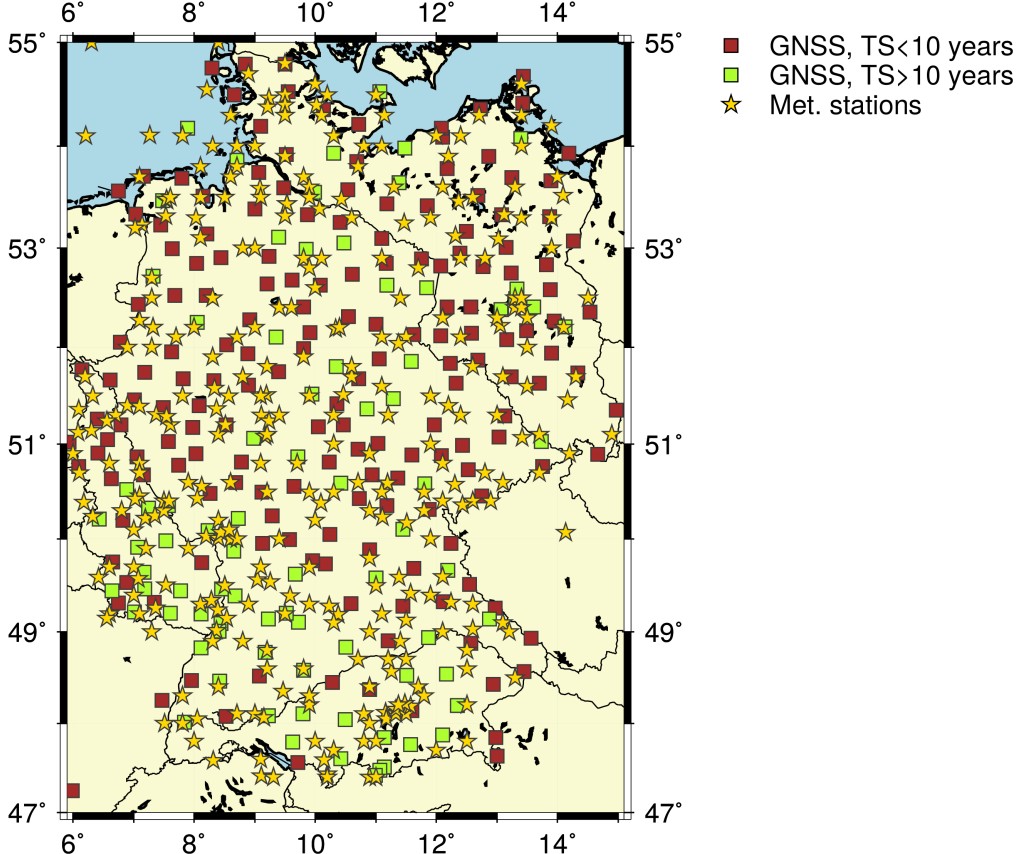

**Figure 1.** The location of the GNSS and meteorological sites within the research region. 119 GNSS sites of 351 have time series of 10 to 19 years long.





| Method | Radiosonde | Dew point-based | ERA-Interim |
|---|---|---|---|
| Least squares | 0.533 | 0.503 | 0.461 |
| Theil-Sen | 0.512 | 0.533 | 0.482 |

**Table 1.** Comparison between the estimated trends (mm/decade) from radiosonde, dew point-based, and ERA-Interim PWV time series at site Lindenberg. The standard error of the estimated trend is $\approx 0.04$ mm/decade.





**Figure 2.** PWV estimated at three GNSS sites (site 0269 in Wertach, Germany at altitude of 907 m AMSL, site 0522 in Pirmasens, Germany at altitude of 399 m AMSL, and site 0285 in Garmisch, Germany at altitude of 1779 m AMSL) and the corresponding PWV from ERA-Interim. The bottom figure shows the mean difference and standard deviation at all sites.





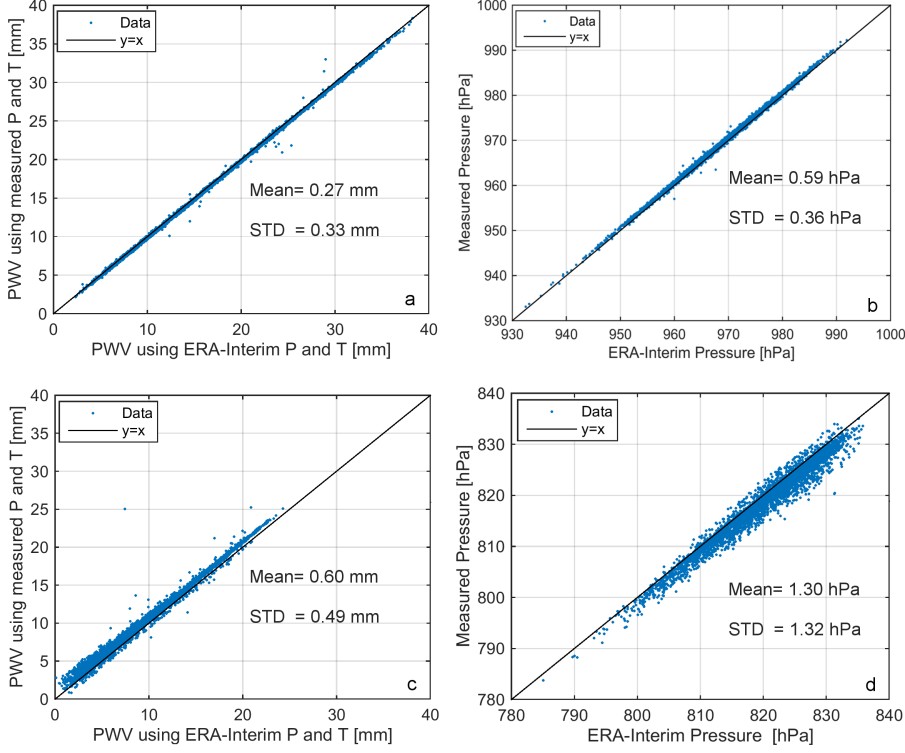

**Figure 3.** (a, c) show PWV determined using interpolated pressure and temperature from surface measurements and ERA-Interim and the corresponding pressure values for the GNSS site 0522. Similarly in (b, d) for the GNSS site 0285.

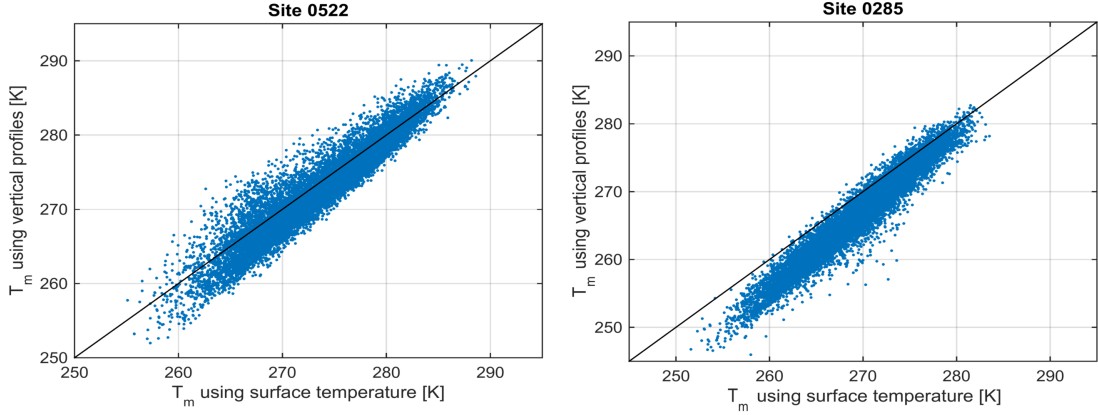

**Figure 4.** Mean atmospheric temperature, $T_m$ determined once using surface temperature and vertical atmospheric profiles from ERA-Interim at the sites 0522 (399 m AMSL) and 0285 (1779 m AMSL). The mean difference is 0.97 K for the first site and 3.02 for the second, and the STD is 2 K for the first and for the second 1.83 K.





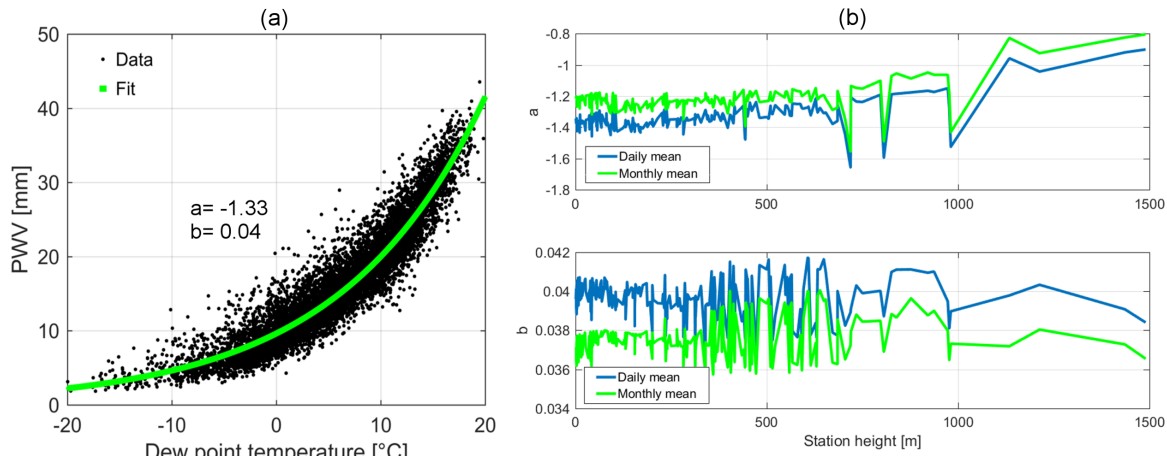

**Figure 5.** The coefficients $a$ and $b$ in Eq. 9 estimated using ERA-Interim PWV data (1991–2016), interpolated at 227 German synoptic stations.

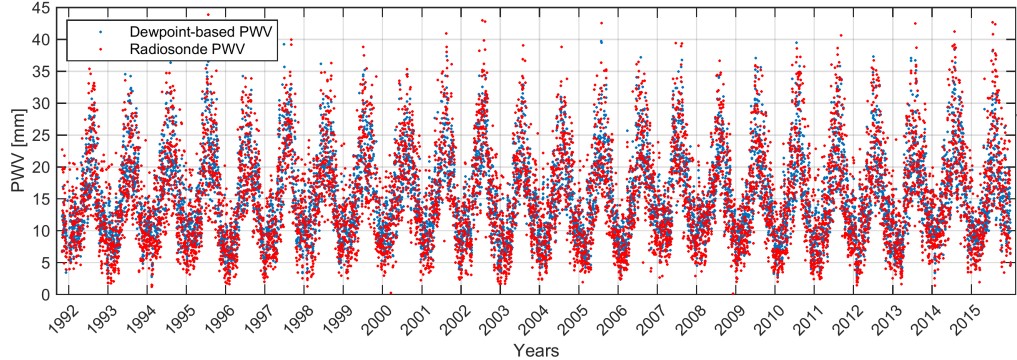

**Figure 6.** Comparison of PWV time series obtained using Eq. 9 and those measured by radiosonde at the site Lindenberg ($14°6'E$, $52°12'N$) at 12:00 UTC. The correlation coefficient is 0.91, mean difference of 0.04 mm, difference STD of 3.2 mm.





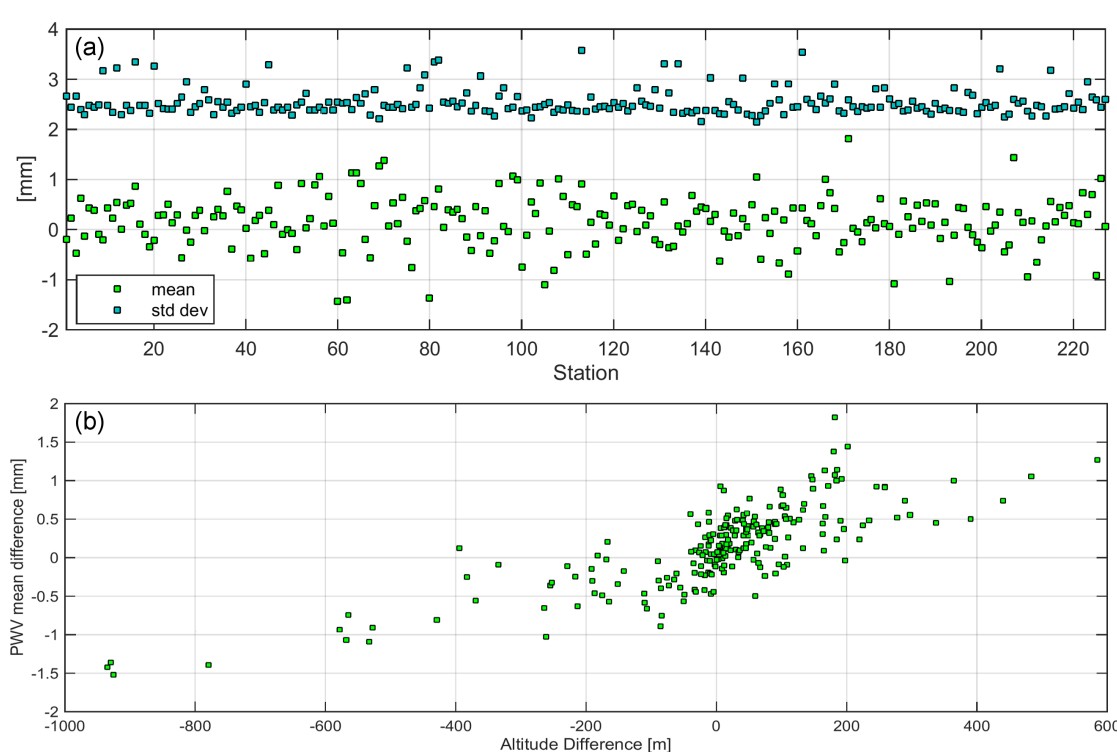

**Figure 7.** (a) Mean and standard deviation of the PWV time series difference (1991–2016) from ERA-Interim and synoptic data at 227 stations. (b) Mean difference against the altitude difference.





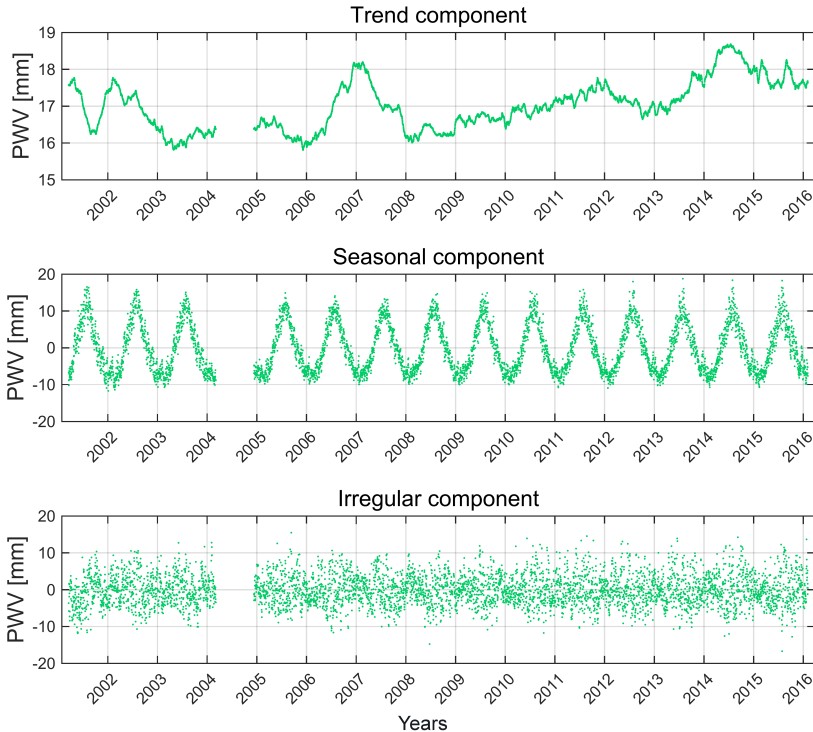

**Figure 8.** Trend, seasonal, and irregular components of PWV time series estimated from GNSS observations (2001–2016) at the site 0896 (Berlin, Germany, 68.37 m AMSL).





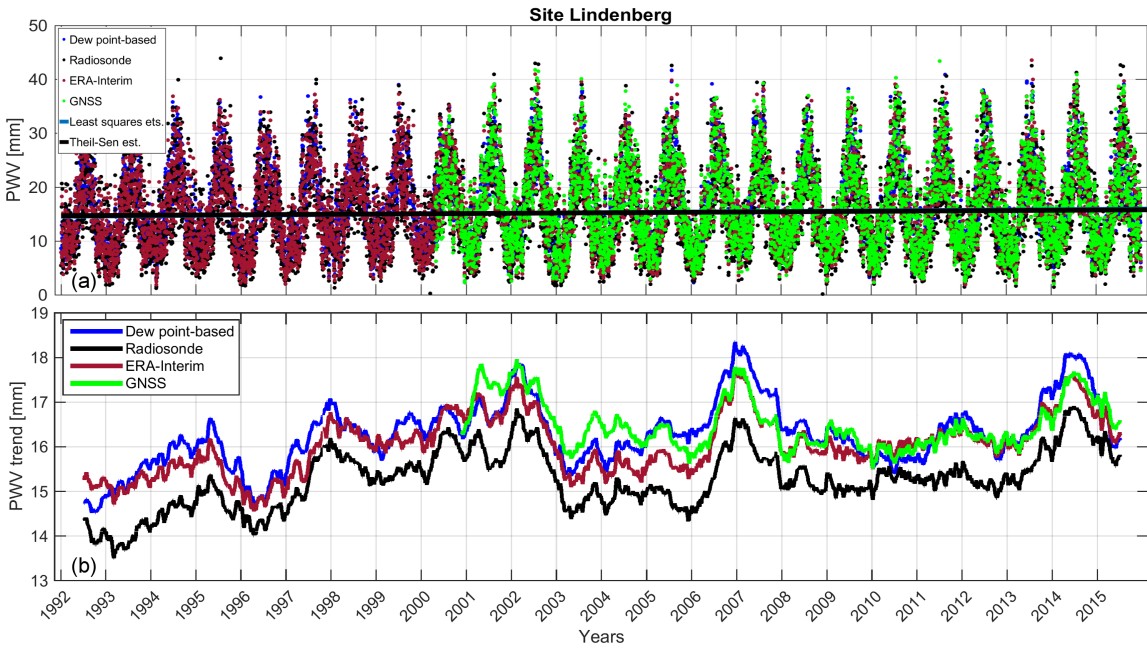

**Figure 9.** (a) Daily mean PWV time series at site Lindenberg from GNSS, ERA-Interim, and synoptic data (1992–2015). The PWV measured by a radiosonde at 12:00 UTC is also shown. The trend extracted by removing the seasonal and irregular components are shown in (b).





**Figure 10.** The estimated PWV trend using GNSS and the corresponding uncertainty in the estimated trend using Theil-Sen estimator. The size of the marker indicates the length of the PWV time series, i.e., the larger the marker, the longer time series. (a, b) show the estimated trends at all sites and the corresponding STD, while (c, d) show the results at sites with times series of at least 10 years length.






**Figure 11.** The estimated PWV trend using (2m) ERA-Interim data by applying least squares regression to the seasonally-adjusted time series (a) and Theil-Sen estimator (c). The standard errors of the estimated trends are shown in (b ,d).



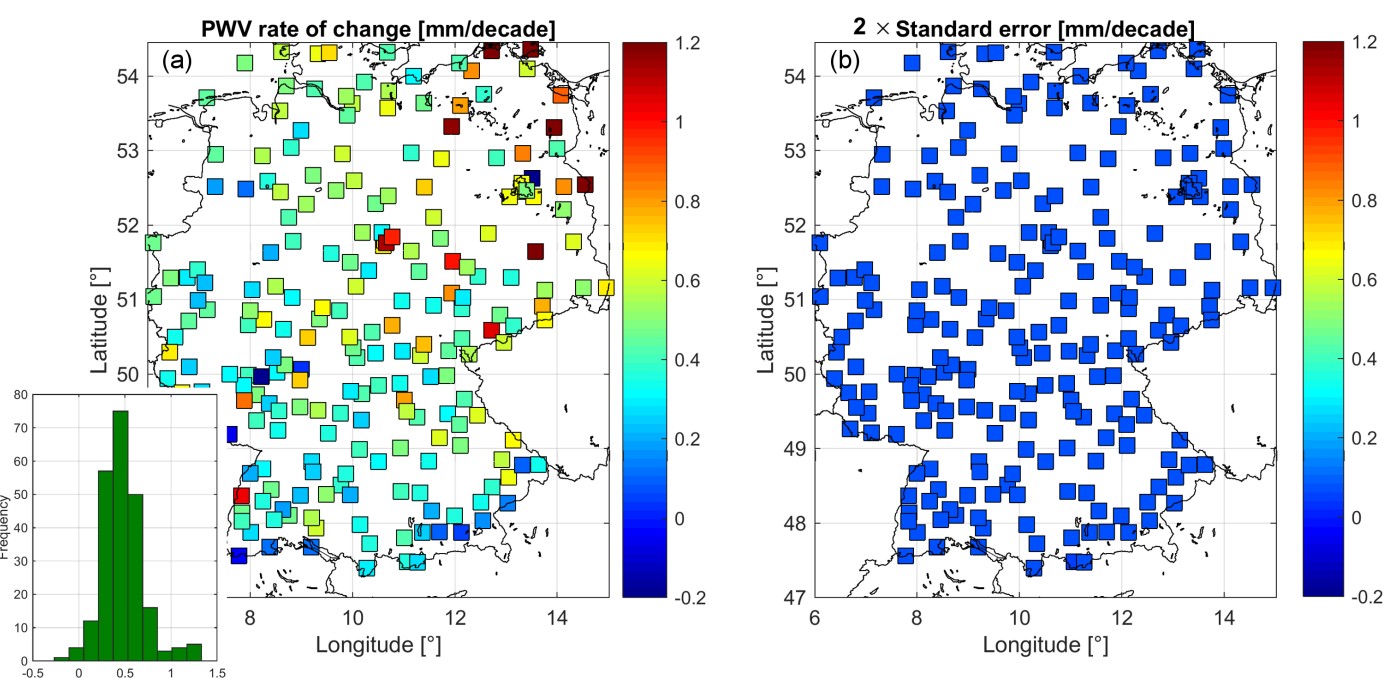

**Figure 12.** (a) Estimated trends using dew point-based PWV at 227 synoptic stations and the corresponding standard deviations of the slope (b).




**Figure 13.** Fitting 3rd degree polynomial to PWV along longitude (a) and to the residuals along latitude (b). The original and the fitted data at 227 stations in (c) are shown on the map in (d ,e).





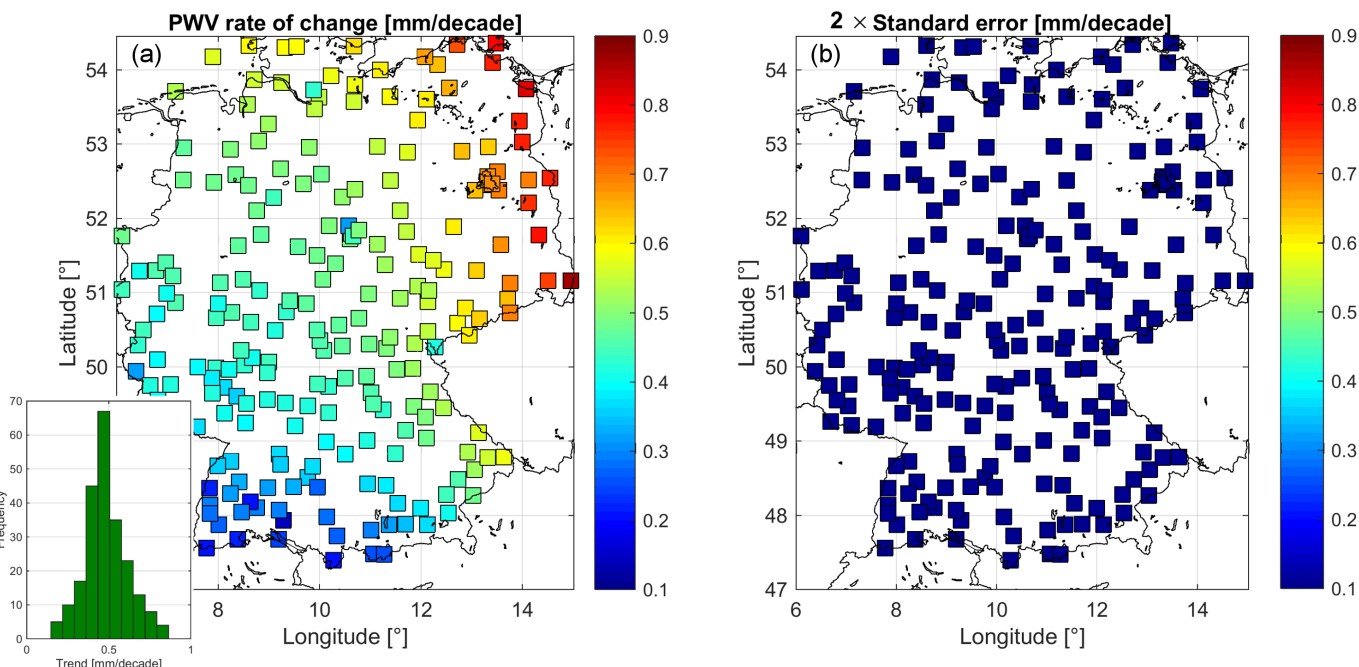

**Figure 14.** Estimated trends using dew point-based PWV after filtering the spatially short scale variations and the corresponding standard error of the estimated slope.



**Figure 15.** Estimated temperature trends using surface measurements (a) before and (c) after spatially short scale variations, and the corresponding standard error of the estimated slope (b, d).





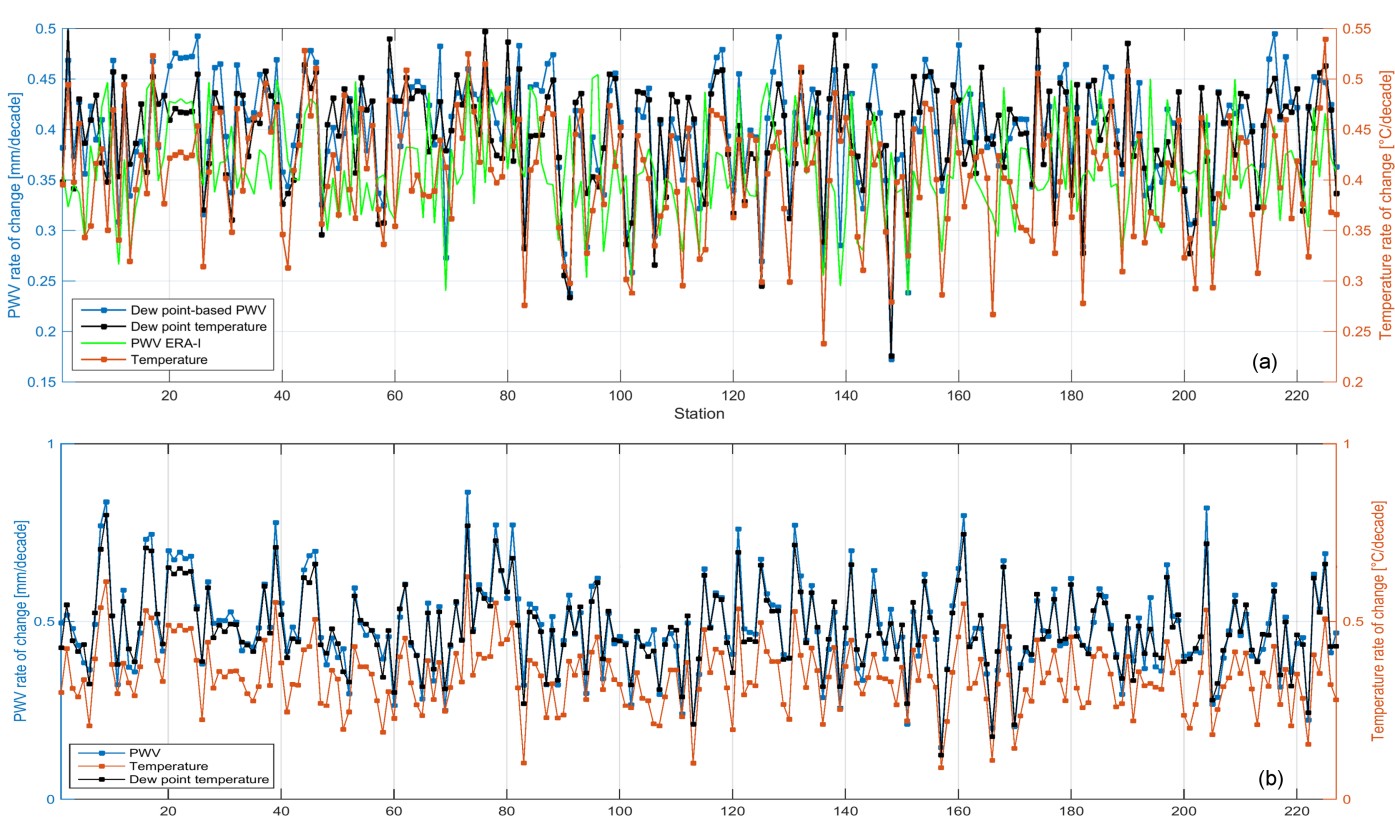

**Figure 16.** Trends of PWV, temperature and dew point temperature from ERA-Interim data (a) and surface measurements (b). The change in PWV is correlated with the change in temperature. The trend of dew point temperature can be considered an as adequate measure for PWV trend.