# Peer review of "Estimating trends in atmospheric water vapor and temperature time series over Germany"

_Atmospheric Measurement Techniques, 2017_

## Referee Comment (RC1) · Anonymous Referee #1 · 13 Apr 2017

The study of Alshawaf et al. (2017) mainly presents an interesting trend analysis of PWV of the GPS station network in Germany. Generally this study is appropriate for AMT. However the presentation could be improved. For example, by reading the abstract and the conclusions it is not so clear which trend values were derived in the study.

Point 1: There is a confusion about what are the main results of this study? Is it the positive trend in Table 1. Is it the map in Figure 10c? Or is it the map in 14c? I would say that Figure 10c is the most important result.

Point 2: I would suggest to give a theoretical explanation for a correlation between a temperature change and a change in water vapor pressure e.g. derived from the Clausius Clapeyron equation. Often one can read that a 1 Kelvin change gives a 7%

change in saturation vapor pressure (Held and Soden, 2000). If the relative humidity is constant then one can derive the expected increase in water vapour pressure too. Does this agree with your results of the trends in T and PWV? Is RH constant over the years?

Held, I. M., and B. J. Soden, 2000: Water vapor feedback and global warming. Annual. Review of Energy and the Environment, 25, 441-475

Point 3: You make a long statement about GPS PWV from mountain regions. I would consider results of this study:

J. Morland, M. Liniger, H. Kunz, I. Balin, S. Nyeki, C. Mätzler, N. Kämpfer: Comparison of GPS and ERA40 IWV in the Alpine region, including correction of GPS observations at Jungfraujoch (3584 m), Journal of Geophysical Research, Atmospheres, vol.: 111, no.: D04102, pp.: 1-12, 2006

Point 4: - Colors in Figure 7 should have more contrast - title in Figure 15 a and c should be T rate instead of PWV rate

Minor corrections:

page 1

line 9: "deseasonalized" is clearer than "seasonally adjusted"

line 13 it is unclear to what "the former" is related. In addition it is unprecise to say the trend is below a certain number. The trend should be equal to a mean value +/- the standard deviation.

page 3:

line 7: There was a study from the Netherlands which showed that PWV has the strongest correlation with the humidity at ca. 1.5 km altitude. Unfortunately I forgot the citation. Such a study demonstrates that surface data cannot be a full substitute for a PWV measurement.

line 20 I would use "finally" instead of "ultimately"

section 4.1 : this method looks a bit unorthodox. I would prefer a trend model and solving the equation system in one step for seasonal oscillation and the linear trend. For equal weighting of the data it would be good to have a complete series of monthly means before analysing the trend.

page 9 line 13 : are available in Figure 9

page 11 line 11 comparison instead of Comparison

line 15 "We found that the trend tends .... It is not clear for me if it is just a result of ERA or if it comes from the GPS measurements too. Figure 10 a and c do not show such a horizontal gradient in the trend

Figure 2 caption please define the "difference". Is it ERA-Interim - GNSS ?

Figure 7 how is the mean difference defined?

Figure 14 and in general: I don't understand why you come up with the dew point-based PWV? Actually you have good and plenty of GNSS data of PWV. Don't you believe in the GNSS PWV trend map , e.g., Figure 10c?

---

## Referee Comment (RC2) · Anonymous Referee #2 · 3 Jul 2017

Comment to: Estimating trends in atmospheric water vapor and temperature time series over Germany, by Alshawaf, Balidakis, Dick, Heise and Wickert, AMT-2017-69

The authors analyse PWV trends in datasets of different origin: Ground-based GNSS, NWP reanalysis, and ground-based weather station data.

The most interesting part of the article concerns the GNSS data, which is also the main field of expertise of the authors. Despite the GNSS time series not being as long as the authors would like, I recommend to put more emphasis on those results, despite the time series not being as long as the authors would have liked.

It is also interesting to see the strong gradient over Germany of the PWV trend in the ERA data, versus no clear gradient in the GNSS data. I recommend to do the ERA

analysis for the same period as covered by GNSS in figure 10 c and d. It is unclear to which degree different time series are included in figure 10 c and d. From figure 8 it is clear that the large variations from year to year of the "trend component", means that differences in time extent risk leading to local variations in figure 10 c. Are there sites enough to do a "clean" figure 10 c, with all sites covering the same period?

Put less emphasis on PWV from ground based meteorological measurements. Even if there is a relation, it is certainly not going to be the way in which we determine PWV variations and trends in the future.

A few more detailed comments

page 2. PWV trends are not similar in all regions, please detail if for example Bengtsson et al cover the same region as you.

page 3 Specify already here the resolution of your vertical ERA profiles (when you finally give the number, you have already used the profile information several times).

page 4, line 13. ..regression -> relation

why not provide eq 6 and 7 already in connection with eq 5?

page 5. When assessing the short commings of finite ERA resolution, why not also check interpolated ERA data directly at the meteorological sites for a clean answer?

page 6 line 21. The standard error of the PWV estimate was deduced against which data?

Figures: In some of the figures PWV differences are shown, but the "sign" is not mentioned. Is it PWV_GNSS - PWV_ERA, or vice versa?

---

## Author Comment (AC1) · 21 Jul 2017

Thank you dear reviewer for these constructive comments and suggestions. Please find our response in the attached pdf file. Best regards.
* * *

---

## Author Response (AR1)

The authors would like to thank the editor and the reviewers for the time they invested to review this paper. We addressed the suggested points. The current version contains all changes according to the reviewer suggestions.

| Referee#1 | Response of the authors |
|---|---|
| Point 1: There is a confusion about what are the main results of this study? Is it the positive trend in Table 1. Is it the map in Figure 10c? Or is it the map in 14c? I would say that Figure 10c is the most important result. | The goal of this research is to derive information about the PWV trends over Germany. The first candidate data set is of course from GNSS. However, two points are important here: first, the length of the time series and the second, the validation of the results. Therefore, other data sets are required. The first is the ERA-Interim. Climatologists suggest estimating the trends in periods of at least thirty years, in order to have reasonable information about the trends. The GNSS data (of interest) are available over different times, starting from 10 years to 19 years. In order to provide a mean value of the trend +/- sigma, the trend at all sites should be estimated over a specific common time window. This is also important for observing spatial features of the trend. For that reason, we chose to estimate the trend over the last 30-year norm defined by the climatologists (1991-2020) or rather as long as the data are available (1991-2016). For that, we used the ERA-Interim data. For validation, we needed a data set, which is of course difficult to find. Therefore, we used the model of the dew-point temperature to obtain the PWV. We know of course that this is not a very precise way to get the PWV, but resources are limited here, and so far, the results from both data sets show an adequate agreement. |
| Point 2: I would suggest to give a theoretical explanation for a correlation between a temperature change and a change in water vapor pressure e.g. derived from the Clausius Clapeyron equation. Often one can read that a 1 Kelvin change gives a 7% change in saturation vapor pressure (Held and Soden, 2000). If the relative humidity is constant then one can derive the expected increase in water vapour pressure too. Does this agree with your results of the trends in T and PWV? Is RH constant over the years? Held, I. M., and B. J. Soden, 2000: Water vapor feedback and global warming. Annual. Review of Energy and the Environment, 25, 441-475 | Thank you. We addressed this point in the current version of the paper by inserting the following: page 8 line 15 to 25, page 10 line 22 to 24, and figure 12. |
| Point 3: You make a long statement about GPS PWV from mountain regions. I would consider results of this study: | Thank you for the suggestion, we added this reference, which agrees with our results. |

| | |
|---|---|
| J. Morland, M. Liniger, H. Kunz, I. Balin, S. Nyeki, C. Mätzler, N. Kämpfer: Comparison of GPS and ERA40 IWV in the Alpine region, including correction of GPS observations at Jungfraujoch (3584 m), Journal of Geophysical Research, Atmospheres, vol.: 111, no.: D04102, pp.: 1-12, 2006 | |
| Point 4: - Colors in Figure 7 should have more contrast - title in Figure 15 a and c should be T rate instead of PWV rate | Modified |
| Minor corrections: page 1 line 9: "deseasonalized" is clearer than "seasonally adjusted" | Modified |
| line 13 it is unclear to what "the former" is related. In addition it is unprecise to say the trend is below a certain number. The trend should be equal to a mean value +/- the standard deviation. | This is a good hint, thank you. However, we do not consider it "right" to put the trend this way for the GNSS data because the trend is estimated for time series of different length. We did that for the other data sets. |
| page 3: line 7: There was a study from the Netherlands which showed that PWV has the strongest correlation with the humidity at ca. 1.5 km altitude. Unfortunately I forgot the citation. Such a study demonstrates that surface data cannot be a full substitute for a PWV measurement. | Yes, we agree with the reviewer on that, and we said in the text that this is not the most accurate way to obtain PWV, but other data sources are limited, particularly when we need long history of data, that is why we relied on this method. Besides, we validated the time series by comparing them with ERA-Interim and radiosonde data (when available) and they showed a high agreement, which makes them suitable for further investigations. We rearranged the paper and modified the text to make this point clear. And according to our analysis, these times series at 227 stations show a very good agreement with the ERA-interim data, so we also used them for trend estimation. |
| line 20 I would use "finally" instead of "ultimately" | Modified |
| section 4.1 : this method looks a bit unorthodox. I would prefer a trend model and solving the equation system in one step for seasonal oscillation and the linear trend. For equal weighting of the data it would be good to have a complete series of monthly means before analysing the trend. | Indeed this method has widely been used in economy, and currently in climate research. Comparing this method with the Theil-Sen estimator assures that it is a proper method for trend estimation. We will consider your suggestion in future research. |
| page 9 line 13 : are available in Figure 9 | Modified |
| page 11 line 11 comparison instead of Comparison | Modified |
| line 15 "We found that the trend tends ... . It is not clear for me if it is just a result of ERA or if it comes from the GPS measurements too. Figure 10 a and c do not show such a horizontal gradient in the trend | It is not possible to look for spatial features of the trend, if it is estimated from time series of different length. Therefore, this is not possible considering the GNSS data since their length varies between 10 and 19 years. This gradient was observed when we analyzed 26-years time series from ERA-Interim and synoptic data. That is why we did not depend on GNSS alone to make conclusions. |

| | |
|---|---|
| Figure 2 caption please define the "difference". Is it ERA-Interim - GNSS ?
Figure 7 how is the mean difference defined? | Yes it is, caption modified (now figure 2).

We meant the mean if the difference between two data sets. Text is modified to make it more understandable. |
| Figure 14 and in general: I don't understand why you come up with the dew point-based PWV? Actually you have good and plenty of GNSS data of PWV. Don't you believe in the GNSS PWV trend map , e.g., Figure 10c? | Thank you for raising the question in this way. We believe in what we are doing, but it seems we need to explain it in a more obvious way to make the idea clear. We explained the narrative as an answer to your first point. The rate of PWV change depending on the starting and ending dates of the time series. In order to give specific conclusions about the trend, we should analyze all stations for a specific period, which should be thirty years according to climatologists. For GNSS, however, the sites are installed independently, so the length of the time series varies within 10-19 years. We still can estimate the trend using these time series. It might, however, be large, e.g. 2.3 mm/decade is a large value of the trend (think of the temperature trend in a similar magnitude), but this value is obtained because the time series is short. If we double the length of the time series, this value will sink. Therefore, we showed that GNSS can be used to provide PWV trends, which was also validated using ERA-Interim data over the same period of time for each stations (Figure 12), however, reasonable values of the trend should be got from longer time series, which why we used the dew point-based PWV and ERA-Interim. |

Response of the authors:

The authors would like to thank the editor and the reviewers for the time they invested to review this paper. We addressed the suggested points. The current version contains all changes according to the reviewer suggestions.

| Referee#2 | Response of the authors |
|---|---|
| ERA data, versus no clear gradient in the GNSS data. I recommend to do the ERA analysis for the same period as covered by GNSS in figure 10 c and d. It is unclear to which degree different time series are included in figure 10 c and d. From figure 8 it is clear that the large variations from year to year of the "trend component", means that differences in time extent risk leading to local variations in figure 10 c. Are there sites enough to do a "clean" figure 10 c, with all sites covering the same period? | Thank you for raising this question. We added figure 9, in which we estimate the trend from concurrent GNSS and ERA-Interim (over the same time window). There is a very good agreement between the two data sets with, expected, slight difference in the trend values. In space, the two data sets behave the same way. |
| Put less emphasis on PWV from ground based meteorological measurements. Even if there is a relation, it is certainly not going to be the way in which we determine PWV variations and trends in the future. | We agree with the reviewer on the point that this is not the best way to obtain PWV. However, external data for validation are limited; therefore, we used these data, which are significantly long and prepared for climate studies. Moreover, we evaluated the time series and found they are suitable for further analysis. Now the paper is rearranged and modified, so that this point is more understandable. |
| A few more detailed comments
page 2. PWV trends are not similar in all regions, please detail if for example Bengtsson et al cover the same region as you. | Yes, that is right. Bengtsson et al. focused on the GPS network in Scandinavian region. |
| page 3 Specify already here the resolution of your vertical ERA profiles (when you finally give the number, you have already used the profile information several times). | Text added (page 3, line 26) |
| page 4, line 13. ..regression -> relation
why not provide eq 6 and 7 already in connection with eq 5? | Modified
Just because we discussed the results of the analysis of the different parameters. |
| page 5. When assessing the short commings of finite ERA resolution, why not also check interpolated ERA data directly at the meteorological sites for a clean answer?
page 6 line 21. | Yes, we also checked the pressure at the meteorological sites obtaining the same results with negligible difference. Since it is important to evaluate the pressure at the GNSS, we added the results at the GNSS sites to the paper rather than the meteorological site. |
| The standard error of the PWV estimate was deduced against which data? | The standard error is conventionally obtained for each data set independently as given in Eq. 15 |
| Figures: In some of the figures PWV differences are shown, but the "sign" is not mentioned. Is it PWV_GNSS - PWV_ERA, or vice versa? | Text added to the caption, it is ERA-Interim−GNSS. |

**Estimating trends in atmospheric water vapor and temperature time series over Germany**

Fadwa Alshawaf[1], Kyriakos Balidakis[2], Galina Dick[1], Stefan Heise[1], and Jens Wickert[1,2]

[1]German Research Centre for Geosciences GFZ, Telegrafenberg, D-14473 Potsdam, Germany.
[2]Technische Universität Berlin, Institute of Geodesy and Geoinformation Science, Straße des 17. Juni 135, 10623 Berlin, Germany.

*Correspondence to:* Fadwa Alshawaf (fadwa.alshawaf@gfz-potsdam.de)

**Abstract.** Ground-based GNSS (Global Navigation Satellite Systems) have efficiently been used since the 1990s as a meteorological observing system. Recently scientists used GNSS time series of precipitable water vapor (PWV) for climate research. In this work, we compare the temporal trends estimated from GNSS time series with those estimated from European Center for Medium-Range Weather Forecasts Reanalysis (ERA-Interim) data and meteorological measurements. We aim at evaluating climate evolution in Germany by monitoring different atmospheric variables such as temperature and PWV. PWV time series were obtained by three methods: 1) estimated from ground-based GNSS observations using the method of precise point positioning, 2) inferred from ERA-Interim reanalysis data, and 3) determined based on daily in situ measurements of temperature and relative humidity. The other relevant atmospheric parameters are available from surface measurements of meteorological stations or derived from ERA-Interim. The trends are estimated using two methods, the first applies least squares to  deseasonalized time series and the second using the Theil-Sen estimator. The trends estimated at 113 GNSS sites, with 10  to 19  years temporal coverage,  vary between -1.5 and  2.3 mm/decade with standard deviations below 0.25 mm/decade. These  results were validated by estimating the trends from ERA-Interim data over the same time windows, which show similar values. These values of the trend depend on the length and the variations of the time series. Therefore, to give a mean value of the PWV trend over Germany, we estimated the  trends using ERA-Interim  spanning from 1991 to 2016 (26 years) at  227 synoptic stations over Germany. The  ERA-Interim data show positive PWV trends  of 0.33±0.06 mm/decade  with standard errors below 0.03 mm/decade.

The increment in PWV varies between 4.5% and 6.5% per degree Celsius rise in temperature, which is comparable to the theoretical rate of Clausius-Clapeyron equation.

[revised manuscript text omitted]
  the bias values below 1 mm and  standard deviation of less than 2 mm (Figure 2). The  bias between the data sets increases for sites in mountainous regions. The time series of the site 0285 (Garmisch, Germany, 1779 m AMSL), for example, show a larger bias between GNSS and ERA-Interim data, which is explained as follows: we average PWV of four distant grid points around the GNSS site. With the rough spatial

10 resolution, the variability of surface topography is not well captured in the reanalysis data, which significantly increases the height difference between GNSS and the model, and hence the PWV difference. Besides, the daily mean in ERA-Interim is obtained by averaging four PWV values/day, while using GNSS there are 96 PWV estimates/day.

15  This positive bias was also observed by *Morland et al.* (2006) when comparing ERA40 and GPS in the Alps for the site Jungfraujoch at 3584 m AMSL.

[revised manuscript text omitted]

5  We also analyze the change in PWV in relation to the change in temperature. As temperature rises, the air capacity to hold moisture increases at the Clausius-Clapeyron rate. The water vapor pressure $e$ is related to temperature $T$ as follows:

$$\frac{e_2}{e_1} = \exp\left(\frac{\Delta H_v}{R}\left(\frac{1}{T_1} - \frac{1}{T_2}\right)\right) \tag{18}$$

where $\Delta H_v$ is enthalpy of vaporization and $R$ is the universal gas constant. This relationship indicates that 1°C rise in the temperature increases the vapor pressure by 7%. Based on this formula, the change in the PWV can theoretically be
10  related to the change in the temperature. The PWV is linearly related to the vapor pressure as presented in (*Tuller*, 1977), i.e., $PWV = 2.3e$. By substituting this into (18), the increment in PWV should, in theory, be the same as the increment in the vapor pressure (approximately 7% ) per degree Celsius rise in temperature. This was also observed by analyzing the temperature, water vapor pressure, and PWV data sets. We obtained the change in PWV and vapor pressure per one degree rise in temperature as shown in Figure 12. The increase in the water vapor pressure at 227 stations is in the range of 4.5% and
15  6.5%, which is close to the Clausius-Clapeyron rate. We observed a similar rate of change for the PWV with the temperature, or more precisely, $\frac{PWV_2}{PWV_1} = 1.003\frac{e_2}{e_1}$.

**4  Results**

**4.1  Estimating the trends using GNSS-based PWV**

In this section, we show the estimated trends using three data sets, GNSS, ERA-Interim, and synoptic data of PWV and
20  temperature. First, we estimated the trends of PWV at 351 GNSS sites with time series of  4 to 19 years long and the corresponding standard deviations of the estimated slope as shown in Figure 7 . The size of the marker is proportional to the length of the time series (small squares indicate short time series and larger ones indicate longer time series). As observed from the figure, there are high trend values, particularly at sites with short time series. Therefore, in Figure  8 (a), we eliminated all sites with time series shorter than 10 years. At the remaining 119 sites the PWV trend varies between -1.5 to
25   2.3 mm/decade (except for six sites) with precision of the estimated trends below 0.25 mm/decade. To validate these estimates, we analyzed ERA-Interim data over the same times where GNSS data are available, Figure 8 (c). The results from concurrent ERA-Interim time series show high similarity in the trend values and the variations of the trend in space.

**4.2  Estimating the trends using longer time series**

Since the trend is estimated from GNSS time series of different length, it is reasonable to provide a mean value for the whole region or observe spatial features of the trends. Therefore, and in order to get more insight and more reasonable conclusions about the long-term temporal variations of PWV, it is necessary to analyze  time series spanning one predefined period for all stations. Since the last climate normal extends from 1991 to 2020, we analyzed time series of 26 years (January, 1991–June, 2016) from ERA-Interim and synoptic data. We investigated time series at 227 meteorological stations where the ERA-Interim is horizontally interpolated at the synoptic station using bilinear interpolation. Figure 9 shows the estimated trends using ERA-Interim PWV time series by applying, first the least squares to the seasonally-adjusted data and second using the Theil-Sen method. Both methods show similar values of the trend, positive with values of $0.34\pm0.06$ mm/decade.  As observed from  Figure 9, the trend tends to increase in the direction to northeastern Germany.

is the length of the time series that might go back to the beginning of the twentieth century. The DWD checks the quality and homogeneity and provides time seriesWe used the surface measurements of temperature~~It is not possible to accurately determine the total column water vapor using surface meteorological observations alone. However, it was shown in the 1960s that it is possible to approximate the atmospheric PWV based on dew point temperature measurements, which is considered an indicator of the amount of moisture in the air (*Reitan*, 1963). The dew point temperature in turn is determined based on the air temperature and relative humidity. *Reitan* (1963) presented a basic relationship between the mean monthly PWV and mean monthly surface dew point temperature by the following regression form:

$$PWV = \exp(bT_d + a) \tag{19}$$

where $PWV$ is in cm and $T_d$ is the dew point temperature in °F. $a$ and ~~compared the results with the ERA-Interim before the trend estimation. The difference between the data sets is displayed in Figure 10. Next, we estimated the trends using the time series of PWV, which are presented in Figure ??. The trends estimated at the sites in the northeastern part of Germany show higher values; however, we do not observe the gradient, with the same consistency, shown by the ERA-Interim data, for example, in Figure 9 (a).Considering these differences, we have to keep in mind that the synoptic data are point measurements that are affected by the local environment (surroundings) of the meteorological station and weather conditions. Also, the parameters~~ $b$ are estimated to have the values of -0.981 and 0.0341 (*Reitan*, 1963). The standard error in the PWV estimate was 0.18 cm. Following the same procedure, *Bolsenga* (1965) obtained slightly different estimates for $a$ and $b$ using hourly and mean daily observations. *Smith* (1966) obtained a similar regression equation with the coefficient $a$ not being constant. It rather depends on the vertical distribution of the atmospheric moisture, i.e.,

$$PWV = \exp\left(\underbrace{0.0393}_{b}T_d + \underbrace{[0.1133 - \ln(\lambda + 1)]}_{a}\right) \tag{20}$$

with the value of $\lambda$ dependent on the site latitude and the season of year (*Smith*, 1966).

In this work, we estimated the coefficients $a$ and $b$ at each meteorological station by fitting the curves in Eq. 19  to the ERA-Interim

5   PWV data. The median values for $a$ and $b$ using daily PWV are -1.346 and 0.039, which are close to the values -1.249 and 0.0427 presented by *Bolsenga* (1965). For monthly PWV, the median values are -1.224 and 0.037 for $a$ and $b$, respectively.

We used measurements of surface dew point temperature to obtain the daily PWV and time series for the whole network are evaluated using the ERA-Interim data. The PWV  value at the meteorological station is computed by applying bilinear

10  interpolation to the ERA-Interim PWV at four grid points around that station. The altitude difference was not accounted for. Figure 10 (a) shows the bias and standard deviation values of daily PWV for 227 ~~stations and the fitted PWV. The fitting is applied in the longitude direction and the fitted PWV is subtracted from the original data and the fitting is applied to the residuals along the latitude. The final fitted PWV is the sum of both fittings along the longitude and latitude. Applying the 1D polynomial regression sequentially over the longitude and latitude leads to better fitting than applying 2D polynomial to the~~

15   stations as well as the bias against the altitude difference of the two data sets (ERA-Interim height−station height). The bias is centered around 0.15 mm and the standard deviation around 2.5 mm. From Figure 10 (b) we observe that the higher the altitude difference, the larger is the mean PWV difference.

Next, we estimated the trends using the time series of dew point-based PWV after removing local environment effects,

20  which are presented in Figure 11. The trend values vary in the range 0.48±0.13 mm/decade over the research region. From the figure, we observe the increase in the estimated trend when moving towards northeastern Germany. The color gradient in this figures is similar to that shown by ERA-Interim in Figure 9. However, the values of the slopes estimated from ERA-Interim and synoptic data are different, which is not surprising. First because of the coarse resolution of the ERA-Interim data and second due to altitude difference, which might result in different trends. In order to justify these results, a data set with a higher

25  spatial resolution than that of ERA-Interim is required.

The same procedure is applied to  estimate the trends from temperature and dew point temperature time series. The estimated temperature trends from surface measurements at 227 stations  shown in Figure  13 (a) fluctuate in the range 0.39±0.1 K/decade. In Figure 13 (c) the trend estimated for dew point temperature time series, in the range 0.48±0.11, are shown. We calculated the change in PWV per one degree rise in temperature, and the results are

30  shown in Figure 12. The increment in PWV is in the range of 4.5% and 6.5%, which is comparable to the theoretical rate of Clausius-Clapeyron equation.

[revised manuscript text omitted]

---

## Author Response (AR2)

Dear Dr. Sussmann,

Thank you very much for your constructive suggestion. We did add the paragraph line 18-28 on page 3 and modified the conclusions.

Best regards on behalf of the co-authors,

F. Alshawaf